# Milk Production of Lacaune Sheep with Different Degrees of Crossing with Manchega Sheep in a Commercial Flock in Spain

**DOI:** 10.3390/ani10030520

**Published:** 2020-03-20

**Authors:** Lizbeth E. Robles Jimenez, Juan C. Angeles Hernandez, Carlos Palacios, José A. Abecia, Anna Naranjo, Jorge Osorio Avalos, Manuel Gonzalez-Ronquillo

**Affiliations:** 1Departamento de Produccion Animal, Facultad de Medicina Veterinaria y Zootecnia, Universidad Autonoma del Estado de México, Campus el Cerrillo, Instituto Literario 100, Toluca, Estado de Mexico 50000, Mexico; lizroblez@hotmail.com; 2Instituto de Ciencias Agropecuarias, Universidad Autonoma del Estado de Hidalgo, Rancho Universitario, Av. Universidad Km 1. Exhacienda de Aquetzalpa, Tulancingo, Hidalgo 43600, Mexico; juan_angeles@uaeh.edu.mx; 3Department of Construction and Agronomy, Faculty of Agrarian and Environmental Sciences, University of Salamanca, Avenida Filiberto Villalobos, 119, 37007 Salamanca, Spain; carlospalacios@usal.es; 4University Institute for Research in Environmental Sciences of Aragon (IUCA), University of Zaragoza, Miguel Servet, 177, 50013 Zaragoza, Spain; alf@unizar.es; 5Department of Animal Science, Animal Nutrition and Environment Modeling Applications Laboratory (ANEMAL), UC Davis, CA 95616, USA; amnaranjo@ucdavis.edu

**Keywords:** Lacaune, Manchega, milk yield, heterosis

## Abstract

**Simple Summary:**

Milk production in Lacaune–Manchega sheep, Robles Jimenez et al. The objective of the present study was to evaluate the effect of the degree of crossbreeding (Lacaune x Manchega) and environmental factors on milk production. Different degrees of purity of the Lacaune breed crossed with Manchega ewes were used as follows: 100% Lacaune, 7/8, 13/16, 3/4, 5/8 and 1/2 Lacaune:Manchega. A mixed model was used to evaluate the level of crossbreeding and environmental factors on milk production. The 100%, 13/16, and 3/4 Lacaune genotypes had the highest milk yields with respect to the 1/2 Lacaune/Manchega breed (*p* < 0.001). It was concluded that 13/16 and 3/4 Lacaune/Manchega ewes presented the highest milk yields with respect to the other crosses.

**Abstract:**

The objective of the present study was to evaluate the effect of the grade of crossbreeding (Lacaune x Manchega) and environmental factors on milk production in a commercial flock in Spain. A total of 5769 milk production records of sheep with different degrees of purity of the Lacaune breed crossed with Manchega were used as follows: 100% Lacaune (*n* = 2960), 7/8 Lacaune (*n* = 502), 13/16 Lacaune (*n* = 306), 3/4 (*n* = 1288), 5/8 Lacaune (*n* = 441) and 1/2 Lacaune: Manchega (*n* = 272). Additional available information included the number of parity (1 to 8), litter size (single or multiple), and the season of the year of lambing (spring, summer, autumn and winter). A mixed model was used to evaluate the level of crossbreeding and environmental factors on milk production. The 100% Lacaune sheep presented the highest milk production with respect to the F1 Lacaune x Manchega sheep (*p* < 0.01), showing that as the degree of gene absorption increases with the Manchega breed, it presents lower milk yield. The 100%, 13/16, and 3/4 Lacaune genotypes had the highest milk yields with respect to the 1/2 Lacaune/Manchega breed (*p* < 0.001). The Lacaune registered on average 181.1 L in a period adjusted to 160 days of lactation (1.13 L/ day). Likewise, the parity number, litter size, and season of lambing effects showed significant differences (*p* < 0.01). It was concluded that 13/16 and 3/4 Lacaune/Manchega ewes presented the highest milk yields with respect to the other crosses.

## 1. Introduction

Sheep milk production is an important economic activity in Spain with a census of 836,216 dairy sheep, which are distributed on 1085 farms [1]. “La Mancha” is a region of Castilla-La Mancha, Spain, that has 37.6% of the dairy sheep farms and contributes about 33% of the sheep´s milk production in Spain. Traditional farms of Manchega sheep are managed under a system known as “mixed sheep-cereal system” which uses natural pastures, crop residues, and stubble that are complementary to an agricultural system of cereal production in dry lands. In recent years, this region has shifted from a mixed sheep-cereal system toward a progressive intensification system [2]. Several modifications to the traditional system were required when exotic breeds that are specialized for milk production were introduced into the national flock.

In the province of Ciudad Real in the Autonomous Community of Castilla la Mancha, Spain, there are currently 335,213 dairy sheep, corresponding to 40% of the total population in the Autonomous Community. The farm studied is located in this region, which initially had 2500 Manchega sheep with a semi-extensive production system. Non-lactating animals graze on communal land or stubble, while animals with lambs or milking animals remain housed as long as possible but are allowed to go out to consume pasture (mainly in spring). The Lacaune dairy breed, which produces the milk for the manufacture of the well-known Roquefort cheese, was introduced into Spain from the 1990s onwards, mainly through artificial insemination (AI) or by importing young animals (seven months old), in order to increase milk production with native Spanish breeds. The breed, which specializes in milk production, is well known for its exceptional production [3]. The aim of the present study was to evaluate the effect of the degree of incorporation of the Lacaune breed in the Lacaune x Manchega crossbreed and environmental factors on milk production of a commercial flock, in Spain.

## 2. Materials and Methods 

### 2.1. Animal Population

The flock consisted of 1600 adult animals for milk production, composed of the Lacaune breed with different degrees of crossbreeding with the Manchega breed. Milk yield records were obtained from the 2003 to 2016 period, using volumetric meters integrated in the milking system to record the milk production of each animal alternating between mornings and afternoons [4]. The data included multiple lactations for some of the animals. Daily milk production was calculated using the method used by the Official Milk Control approved by the International Committee for Animal Recording [4]. Considering the time between each milking, using the formula as follows: Daily milk = (Registered milk * 24)/(Time between milk records)

Milk production per month and total lactation were calculated using the International Committee for Animal Recording (ICAR) method, 2016, with alternating monthly controls [4].

Animals were raised in an intensive production system during the lactation period. They did not go outside and were fed in the installation with a unifeed mixture of forage and concentrates, established by differentiating three qualities, according to the lactation period as follows: high production (first two months postpartum), medium production (three months later), and low production at the end of the lactation period (last two months). Food of similar quality and quantity was offered to the ewes. Animals were mated five months postpartum, allowing some ewes to lamb twice in the same year. The production system aims for the majority of sheep to lamb on three occasions in two years, which is similar to the STAR^©^ accelerated lambing system in sheep [5]. 

### 2.2. Population Records

The database included 5769 lactating sheep with varying percentages of Lacaune crossbreeding (Table 1). The parental line was determined with father line (FL) or mother line (ML) crossbreeding, using AI or natural mating (NM) rams, starting with 100% Lacaune sheep (*n* = 2960), 7/8 Lacaune + 1/8 Manchega (*n* = 502), 13/16 Lacaune + 3/16 Manchega (*n* = 306), 3/4 Lacaune + 1/4 Manchega (*n* = 1288, 5/8 Lacaune + 3/8 Manchega (*n* = 441), and 1/2 Lacaune + 1/2 Manchega (*n* = 272). 

The 100% Lacaune sheep were acquired, with their pedigree from France. The Manchega sheep breed were acquired with pedigree registered in AGRAMA’s (National Association of Manchega Sheep Breeders) genealogical book. Subsequently, the crosses (AI or NM) were made in a visually controlled manner with Lacaune rams, in addition to enhancing maternity and paternity controls with DNA tests; this type of test is based on the analysis of highly polymorphic DNA markers, for which microsatellite markers, also known as short tandem repeat (STR), were used.

Data collected included the percentage of Lacaune, interval between parities, days dry, date of lambing, lactation length, milk production per period, and parity number and litter size (single or multiple).

### 2.3. Editing Information

The descriptive statistics of the final database are shown in Table 2. 

Due to the structure of the data, parities from 8 to 13 were merged to only one category called >8, therefore eight levels were obtained (1 to 8). For litter size, data of triple lambing (29 births) were grouped with the double-birth data, therefore the following two levels were considered: single birth (*n* = 4011) and multiple (twins and triplets, *n* = 1758). Likewise, milk yields were analyzed during the seasons of the year (spring, summer, autumn and winter). The structured database in Excel was transformed into text files with the Texpad program. The total milk yield (TMY) was fitted to 160 days of lactation with the aim of homogenizing the database.

### 2.4. Statistical Analysis

A mixed model was used to evaluate the effect of percentage of Lacaune breeding and environmental factors on milk production. The effect of lactation days (linear and quadratic) as a covariate, used in previous studies [6], was considered. The model is the following:Y*_ijklm_* = μ + Npl*_i_* + Np*_j_* + Tp*_k_* + Ep*_l_* + β1(x_ijklm_ − x) + β2(x^2^_ijklm_ − x^2^) + O*_m(l_*_)_ + e*_ijklm_*
where Y*_ijklm_* = milk production (l), μ = overall mean., Npl*_i_* = fixed effect of the i-th level of the proportion of Lacaune breeding (*n* = 6), Np*_j_* = fixed effect of the j-th level of the number of lambing factor (*n* = 8), Tp*_k_* = fixed effect of the k-th level of the type of lambing factor (*n* = 2), Ep*_l_* = fixed effect of the l-th level of season of lambing factor (*n* = 4), β1 (*_xijklm_*) = coefficient of linear regression of the covariate lactation days, β2 (*_ijklm_*) = coefficient of quadratic regression of the covariate lactation days, O*_m_* = random effect of the m-th sheep nested within the i-th (*n* = 5769), and e*_ijklm_* = random error.

The restricted maximum likelihood (REML) method was used to fit the proposed linear model. All the analytical procedures were implemented in the ASReml software (v3.0.5) [7], to estimate the minimum square means. For the multiple comparisons of means, evaluating the significance with the Holm method [8], it was, therefore, appropriate to compare in an adequate way minimum quadratic means obtained from a mixed model. Pirate plots were drawn using the Yarrr package [9,10]. 

The lactation curves of each genetic group were analyzed using the Wood incomplete gamma function (WD) [11]. The WD model is expressed as:Y = at^b^*e*^−ct^
where a, b, and c are the parameters that describe the shape of the curve. The parameter a represents the milk yield at the beginning of lactation; b and c are the parameters that describe the inclining and declining slopes of the lactation curve. The Wood parameters were estimated through the iterative nonlinear curve fitting procedure of regression analysis using the “nls” function in the R software [10]. These parameters were used to estimate the peak yield (PY), time at peak yield (TPY), and persistence (PER), which maintains a higher level of milk production for a longer time during lactation. 

## 3. Results

### 3.1. Milk Production in The Lacaune Breed and Its Crosses

Milk production is affected by different factors, such as those linked to the animal (intrinsic) and those linked to the environment and management (extrinsic), as well as the interaction between factors. Table 2 shows the descriptive statistics of the interval variables between births, milking days, and total milk production for the different degrees of purity of the Lacaune x Manchega dairy sheep used in this study.

In Table 3 and Figure 1, milk production is presented, where the 100%, 13/16, and 3/4 Lacaune sheep had the highest milk production (181 L in 160 day of lactation) with respect to the other genotypes (cross F1 and 1/2 Lacaune x 1/2 Manchega) (*p* < 0.01).

Table 4 and Figure 2 show the lactation curves, where the 100% Lacaune genotype obtained the highest PY (1.69), and the genotypes with shorter TPY were 7/8 Lacaune + 1/8 Manchega, followed by 13/16 Lacaune + 3/16 Manchega and 3/4 Lacaune + 1/4 Manchega (*p* < 0.001).

### 3.2. Effect of The Parity Number, Litter Size, and Season of Lambing

Animals showed the lowest milk yield during the first lactation (*p* < 0.01), and increased linearly in the second and third ones, maintaining and reaching the maximum production until the fifth lactation, before production decreased at the sixth lactation (*p* < 0.01) (Table 1). In the same way, ewes with litter sizes of two or more presented the greatest milk production (*p* < 0.01). The highest milk yields were obtained in winter (*p* < 0.01) (Figure 3). 

## 4. Discussion

### 4.1. Milk Production of the Lacaune Breed and Its Crosses

The Lacaune is one of the breeds with the highest milk yield, efficiently selected for milk yield and composition, improving yields year after year due to the establishment of a genetic selection scheme. There are few available comparisons of crosses with the Lacaune breed, except in Spain, where crosses with the Churra breed have been made [3].

The 100% Lacaune sheep produced the highest milk yields with respect to F1 Lacaune x Manchega sheep (*p* < 0.01). Figure 1 shows that as the degree of purity of the Lacaune breed decreased, milk production also decreased. The findings of other authors who evaluated crosses with Lacaune agree with this study, finding that milk production decreased in Lacaune sheep crosses with other sheep breeds [12].

Konečná et al. [12] also mentioned that when crossing Lacaune and East Friesian, they obtained better nutritional characteristics of the milk in F1 sheep, with increased protein and casein. 

Since a commercial farm was studied, individual milk chemical composition was not determined. 

Thomas et al. [13] and Ángeles-Hernández et al. [14] mentioned that increasing the purity of some breeds of dairy sheep increased the presence of respiratory diseases. Furthermore, in the meat production industry it has been found that the greater the degree of heterosis, the higher the productive yields [15,16]. The findings of this study indicated that heterosis was not found to be significantly important in the crossing of the Lacaune x Manchega breed for milk production.

In this study, Lacaune sheep produced 181.1 L in a period adjusted to 160 days of lactation (1.13 ± 0.31 L per day, with a coefficient of variation of 27.24%), similar to milk yields in the Slovak Republic of 1.05 L/day [6]. The latter study also highlighted the effect of the flock-test-day of test on the variability of the daily milk production, whereas, in our study, this was not possible to obtain, because those were data corresponding to a single flock, but our study provided reasonable and valuable results on the performance of milk in this breed under the conditions of a commercial flock. 

Thomas et al. [17] found that the Lacaune breed had milk yields on average of 1.64 l/day. Perhaps the difference was due to their study being conducted in a research station in the USA, under more controlled conditions.

Thomas et al. [17] crossed Lacaune x East Friesian (EF) and found average daily yields of 0.98 and 0.92 l/day in sheep with genotypes 3/4 and 1/2 x EF, respectively. Rovai et al. [18] found that the Lacaune breed showed yields of 1.36 kg per day, which were higher than the present study.

In relation to the lactation curves (Table 4 and Figure 2), the Lacaune sheep showed a higher lactation PY (1.69 and *p* = 0.002); however, it showed a lower persistence (the maintenance of a higher level of milk production for a longer time during lactation) (4.23 and *p* = 0.001); PY and PER together define the shape of the lactation curve and area under the curve, thus the total milk yield per lactation. Genetic groups with higher TPY (13/16 Lacaune + 3/16 Manchega and 3/4 Lacaune + 1/4 Manchega) showed lower values of PY, but higher levels of PER, which is desirable since greater genetic variability has been shown to reduce the likelihood of dairy sheep suffering metabolic stress [19]. In addition, the selection of animals with higher PER could increase TMY without increasing the occurrence of reproductive failure [20].

Ugarte et al. [21] mentioned that the average estimated milk production in a set of farms of the Manchega breed was between 90 and 100 L per animal per year, although this disagrees with the data produced by farmers belonging to AGRAMA, confirming the existence of great variability of production and the potential that exists, using data corresponding to the Official Milk Control (1.2 kg per day and 120 kg per lactation at 100 days) [22].

There are also important differences in milk production with other studies. Productive yields have been reported in the Lacaune breed with an average production of 320 kg per lactation, average lactation of 171 days of milking with 247 kg of normalized milk production [1]. Barillet et al. [3] found that the production of the Lacaune, in the Roquefort region, France, was from 200 to 270 L in 165 days of lactation. This could be due to the differences between production levels and objectives, as well as the efforts to improve the production of the Lacaune breed in its country of origin, and since that time, complete lactations with an average production of 250 to 300 L have been reported [3]. It is known that the most important factor affecting the productive variables is the livestock production system (i.e., grazing pastures or feed supplementation with concentrates).

### 4.2. Effect of Parity Number, Litter Size, and Season of Lambing

Sheep in their fifth lactation reached maximum milk production (*p* < 0.01) (Table 3). Rivasa et al. [2] found similar results to this study, where the effect of parity number had significant differences (*p* < 0.01), with sheep from the second to the seventh parity showing the highest milk yields with respect to those sheep that recorded births past the eighth parity. Similarly, Rovai et al. [18] also observed that the first parity corresponded with the lowest yields (*p* < 0.01), as well as in other studies conducted in Spain [23,24].

Prolificacy positively influences dairy milk production, being higher in those sheep that gestated two or more lambs as compared with sheep that only birthed one lamb [16,24] (Figure 3). This difference is determined by the total volume of the placenta and an increased production of placental hormones (galactono-placental) that influence the development of the mammary structure before birth. There is also a difference due to the sex of the offspring [18,23,24]. Thus, this effect must be differentiated from the stimulus of lambs that subsequently occurs during the suckling [18]. It has been proven that sheep that birthed two lambs and raised one produced 30% to 40% more milk as compared with those that raised and reared a single lamb [25]. These results agree with the present study, although there were no pronounced differences, given that multiple-birth ewes showed 7% higher milk yield than those in the single birth group (*p* < 0.01); very similar results were found with sheep of the Assaf and Castellana breeds, finding 8% and 9% higher milk production, respectively, when the sheep had multiple births [23,24].

The season of lambing has a complex effect on the total milk production of the sheep, from the availability in the pastures in the grazing systems, but also through environmental factors, the effect of temperature, humidity, light hours, and even the farming systems (feedlot or grazing). Caja et al. [25] and Peterson et al. [26] reported that lactations that started in the spring instead of autumn, reached a higher lactation peak (the state of lactation and the time of year acting synergistically). The environmental conditions that were present in this study had an influence on milk production yields (Figure 3); given the lambing season’s significant differences, the productive milk yields were higher in the winter and spring, while lower in summer and autumn, as was also found in previous studies with representative breeds (Castellana and Assaf) in Spain [23,24]. These results are similar to those of Rovai et al. [18] who also found differences in milk production depending on the season of the year. However, Rovani et al. [18] found lower milk yields in winter, while our study did not see lower yields in winter, but instead saw lower yields in summer and autumn.

## 5. Conclusions

The pure Lacaune sheep produced the highest milk yield, indicating that as the degree of purity of the Lacaune decreased with the Manchega cross breeding, milk yield decreased. Lacaune and Manchega crosses 13/16 and 3/4 presented the highest milk yields with respect to the other crosses.

## Figures and Tables

**Figure 1 animals-10-00520-f001:**
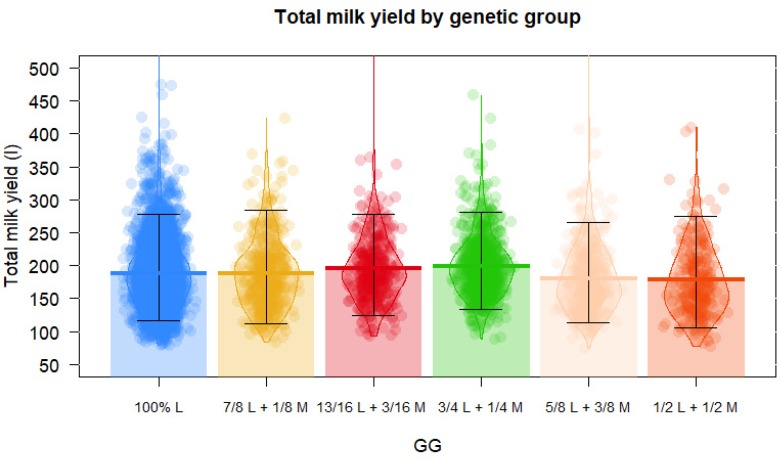
Pirate plot of the effect of genetic group on total milk yield of dairy sheep from a commercial flock in Spain. (GG, genetic group; L, Lacaune; M, Manchega crossbreeding with Lacaune; points represent the raw data; bar/line is the descriptive statistic (mean); bean is the smoothed density curve showing the full data distribution; and brackets represent the confidence intervals).

**Figure 2 animals-10-00520-f002:**
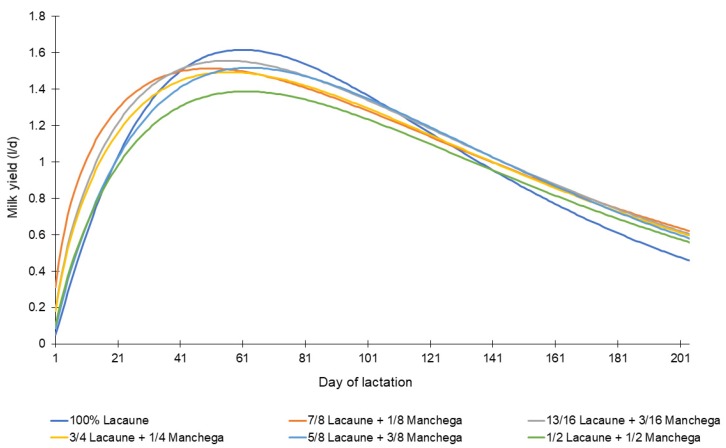
Lactation curves (fitted with the Wood gamma model) of Lacaune and their crosses at different degrees with Manchega dairy sheep from a commercial flock in Spain.

**Figure 3 animals-10-00520-f003:**
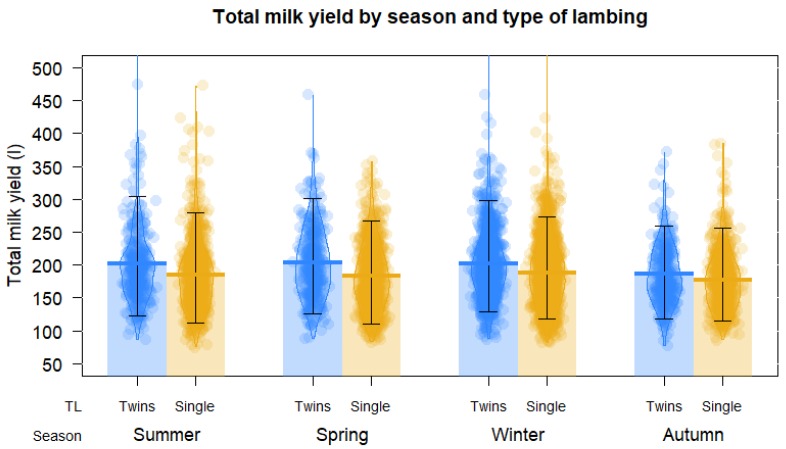
Pirate plot of the effect of litter size (TL) and season of lambing on total milk yield of dairy sheeep from a commercial flock in Spain (points represent the raw data; bar/line is the descriptive statistic (mean); bean is the smoothed density curve showing the full data distribution; and brackets represent the confidence intervals).

**Table 1 animals-10-00520-t001:** Description of the dairy sheep Lacaune (L) with Manchega (M) crossbreeding; parental line was determined with father line (FL) or mother line (ML) crossbreeding, using artificial insemination (AI) or natural mating (NM) rams.

Genetic Group	*n*	Father Line	Mother Line	Mating
100% Lacaune	2960	Lacaune	Lacaune	NM
7/8 Lacaune + 1/8 Manchega	502	Lacaune	3/4L + 1/4M	AI
13/16 Lacaune + 3/16 Manchega	306	5/8L + 3/8M	5/8L + 3/8M	NM
3/4 Lacaune + 1/4 Manchega	1288	Lacaune	1/2L + 1/2M	AI
5/8 Lacaune + 3/8 Manchega	441	3/4L + 1/4M	7/8 L + 1/8M	NM
1/2 Lacaune + 1/2 Manchega	272	Lacaune	Manchega	AI

L, Lacaune; M, Manchega.

**Table 2 animals-10-00520-t002:** Global descriptive statistics of the different degrees of crossbreeding of Lacaune x Manchega dairy sheep in a commercial flock, in Spain.

Variable	Mean ± SD	Min–Max	C.V. (%)
Interval between births (days)	123.8 ± 88.7	63.2–201.6	70.3
Drying-lambing (days)	63.1 ± 56.9	13.7–187.5	90.3
Milking days	161.6 ± 44.4	79.8–270.7	27.4
Total milk yield (L)	176.1 ± 81.5	90–382	46.2
Daily milk production (L/day)	1.064 ± 0.367	0.10–3.55	34.5

C.V. = coefficient of variation; Interval between births (days) is the interval between successive lambing; Drying–lambing (days) is the time between the ewe drying up to the start of the next parity lactation period.

**Table 3 animals-10-00520-t003:** Least square means and standard error of the mean of the degrees of purity of the Lacaune breed in daily and total milk production adjusted 160 days, as well as the parity number, litter size, and lambing season, as environmental effects.

Genetic Group	*N*	L/day	Total Milk Yield
100% Lacaune	2960	1.07 ± 0.01 ^ab^	181.09 ± 1.62 ^a^
7/8 Lacaune + 1/8 Manchega	502	1.07 ± 0.02 ^ab^	179.46 ± 2.74 ^ab^
13/16 Lacaune + 3/16 Manchega	306	1.15 ± 0.02 ^a^	185.19 ± 3.27 ^a^
3/4 Lacaune + 1/4 Manchega	1288	1.08 ± 0.01 ^ab^	181.04 ± 1.93 ^a^
5/8 Lacaune + 3/8 Manchega	441	1.04 ± 0.02 ^ab^	175.43 ± 2.78 ^ab^
1/2 Lacaune + 1/2 Manchega	272	1.02 ± 0.02 ^b^	169.80 ± 3.39 ^b^
Total	5769	*p* = 0.02	*p* = 0.001
**Parity Number**			
1	1232	0.99 ± 0.01 ^c^	165.22 ± 2.25 ^c^
2	1268	1.11 ± 0.01 ^ab^	183.58 ± 2.21 ^a^
3	1057	1.11 ± 0.01 ^ab^	184.61 ± 2.19 ^a^
4	829	1.12 ± 0.01 ^a^	183.88 ± 2.28 ^a^
5	580	1.07 ± 0.02 ^ab^	182.08 ± 2.56 ^a^
6	380	1.05 ± 0.02 ^bc^	178.65 ± 2.99 ^b^
7	230	1.05 ± 0.02 ^bc^	178.06 ± 3.68 ^b^
≥ 8	193	1.03 ± 0.03 ^bc^	173.27 ± 3.99 ^b^
Total	5769	*p* = 0.001	*p* = 0.001
**Litter Size**			
Single	4011	1.02 ± 0.01 ^b^	172.65 ± 1.67 ^b^
Multiple	1758	1.11 ± 0.01 ^a^	184.69 ± 1.85 ^a^
Total	5769	*p* = 0.001	*p* = 0.001
**Season of Lambing**			
Spring	1647	1.10 ± 0.01 ^a^	182.27 ± 2.00 ^a^
Summer	1150	1.07 ± 0.01 ^a^	164.00 ± 2.39 ^b^
Autumn	987	1.02 ± 0.01 ^b^	166.55 ± 2.58 ^b^
Winter	1985	1.08 ± 0.01 ^a^	182.17 ± 1.82 ^a^
Total	5769	*p* = 0.001	*p* = 0.001

^abc^ Different letters in the same column are different statistically (*p* < 0.001). Parity number, due to some animals lambing more than once a year, some animals have more parity records; additional parities above 8 were grouped as ≥8.

**Table 4 animals-10-00520-t004:** Mean parameter estimates of Wood gamma model and characteristics of lactation curves according to the genetic group (Lacaune and their crosses at different degrees with Machega dairy sheep).

Genetic group	A	B	c	PY	TPY	Persistence
100% Lacaune	0.053 ^i^	1.11 ^d^	0.018 ^d^	1.69 ^d^	61.66 ^e^	4.23 ^g^
7/8 Lacaune + 1/8 Manchega	0.315 ^d^	0.53 ^h^	0.011 ^e^	1.44 ^f^	48.18 ^g^	5.93 ^d^
13/16 Lacaune + 3/16 Manchega	0.188 ^e^	0.69 ^g^	0.012 ^e^	1.54 ^e^	57.51 ^f^	5.25 ^e^
3/4 Lacaune + 1/4 Manchega	0.183 ^f^	0.69 ^g^	0.012 ^e^	1.50 ^e^	57.49 ^f^	5.23 ^e^
5/8 Lacaune + 3/8 Manchega	0.087 ^h^	0.91 ^e^	0.014 ^e^	1.56 ^e^	65.0 ^d^	4.64 ^f^
1/2 Lacaune + 1/2 Manchega	0.106 ^g^	0.82 ^f^	0.013 ^e^	1.39 ^g^	63.07 ^e^	4.80 ^f^
S.E.	0.0014	0.007	0.038	0.013	0.51	0.043
*p*-Value	0.001	0.01	0.01	0.002	0.001	0.001

a, b, and c are the Wood gamma model’s parameters; PY, peak yield lactation (L/d); TPY, time at peak lactation (days). ^defghi^ Different letters in the same column are statistically different (*p* < 0.01).

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
