# Peer review of "Milk Production of Lacaune Sheep with Different Degrees of Crossing with Manchega Sheep in a Commercial Flock in Spain"

_animals, 2020, doi:10.3390/ani10030520_

Round 1

Reviewer 1 Report

English language and grammar needs some editing throughout the manuscript to improve readability.

Page

Line

Comment

2

72 and 81

What is the difference between these numbers of animals?

8

Table 4

In the results the authors state where some of these parameter estimates are different between genetic groups, but what is the basis for stating that they are different?  Is there a significance level associated with each estimate that indicates differences?  How is it determined?

12

205-206

I think the authors may have referenced the incorrect Figures for this data they are presenting here.

220-225

This is a very long sentence and is difficult to follow the information being presented.  It should be broken up into multiple sentences for better reading.

226-229

This is a very long sentence and is difficult to follow the information being presented.  It should be broken up into multiple sentences for better reading.

229-232

How can the authors state production levels of the purebred Manchega sheep when they do not have them in their study?

234

It would help to define the term “Persistence” s it is used here.  Does it mean that the ewes maintain a higher level of milk production for  a longer time during lactation?

236

Define the term “TMY”

241-245

This is one long sentence and a paragraph should not be just one sentence. It should be broken up into multiple sentences for better reading.

13

250-253

This sentence is confusing.  Needs to be reworded.

276

Should it be “Table 3” instead of “Figure 3”?

Author Response

Manuscript ID    animals-690517

Type   Article

Number of Pages   16

Title

Milk production of Lacaune sheep with different crossing degrees with Manchega sheep in a commercial flock in Spain

Authors

Lizbeth Esmeralda Robles Jimenez , Juan Carlos Angeles-Hernandez , Carlos Palacios Riocerezo , Alfonso Abecia , Anna Naranjo , Jorge Osorio Avalos * , Manuel Gonzalez Ronquillo *

Dear  Editor  and   Reviewers,

I would like to thank you for the opportunity of letting us improve and resubmit the manuscript. We believe that your suggestions have identified areas to improve that will benefit the overall presentation and content of the manuscript. Attached you will find the updated manuscript version, with the following modifications that have been applied.

Please note that although changes have been highlighted in red font, deletions of content could not be indicated

Sincerely  the authors

Comments reviewer 1

English language and grammar needs some editing throughout the manuscript to improve readability.

R Manuscript has been revised by a native English speaker

Page

Line

Comment

2

72 and 81

What is the difference between these numbers of animals?

R The herd consists of 1,600 adult animals for milk production, composed of the Lacaune breed with different degrees of crossbreeding with the Manchega breed. The data of some animals had more than one lactation, for these reason the  total database included 5,769 sheep lactations with varying percentages of Lacaune crossbreeding (Table 1).

8

Table 4

In the results the authors state where some of these parameter estimates are different between genetic groups, but what is the basis for stating that they are different?  Is there a significance level associated with each estimate that indicates differences?  How is it determined?

R: We are according with the comment. Support of better way this point was carried out a ANOVA between genetic groups; where the Wood´model parameters (a, b and c), PY, TPY and Per were the dependent variables and the explanatory variable was the genetic group. Post hoc analysis was effectuated using a Tukey test. The significance test were added in Table 4.

12

205-206

Q I think the authors may have referenced the incorrect Figures for this data they are presenting here.

R  Has been corrected

220-225

Q This is a very long sentence and is difficult to follow the information being presented.  It should be broken up into multiple sentences for better reading.

R  has been  divided in  short sentences

226-229

This is a very long sentence and is difficult to follow the information being presented.  It should be broken up into multiple sentences for better reading.

R  has been  divided in  short sentences

229-232

How can the authors state production levels of the purebred Manchega sheep when they do not have them in their study?

R There is a misunderstanding, the   data showed in the paper referred to another author, in order to compare with the present study,  sentence has been deleted in order  to  avoid  confusion.

234

It would help to define the term “Persistence” s it is used here.  Does it mean that the ewes maintain a higher level of milk production for  a longer time during lactation?

R, Yes you are  right. L146-147

Persistence (which maintains a higher level of milk production for a longer time during lactation).

236

Define the term “TMY” 

R has been defined as Total milk yield   L118 

241-245

This is one long sentence and a paragraph should not be just one sentence. It should be broken up into multiple sentences for better reading.

R  has been  divided in  short sentences

13

250-253

This sentence is confusing.  Needs to be reworded.

R Sentence has been rewriting

276

Should it be “Table 3” instead of “Figure 3”?

R Has been corrected

Reviewer 2 Report

The paper has to be integrated with information about breeding and mainly feeding system. In addition,  milk yield without milk chemical composition has not complete scientific soundness. In my opinion the paper should be revised taking into account these suggestions. 

Author Response

Reviewer 2

Q The paper has to be integrated with information about breeding and mainly feeding system.

R L84-91 Animals were raised in an intensive production system during the lactation period. They did not go outside and were fed in the installation with a unifeed mixture with forage and concentrates, established by differentiating three qualities, according to the lactation period: high production (first two months postpartum), medium production (three months later) and low production at the end of the lactation period (last two months). No racial differences were made in the quality or quantity of the food offered to the ewes. The animals were mated after five months postpartum, so every year there are animals that can give birth twice in the same year. The productive system tries to make the majority of sheep produce three deliveries in two years, similar to the Star system of meat sheep.

Q In addition, milk yield without milk chemical composition has not complete scientific soundness. In my opinion the paper should be revised taking into account these suggestions.

 R.L232-233 Since a commercial farm was studied, there were a limitation on individual dairy controls, therefore milk chemical composition was not studied due to the lack of data.

Round 2

Reviewer 2 Report

The authors addressed the reviewers suggestions. Thus the manuscript is now acceptable. 

Author Response

Ms. Ref. No.: animals-690517

Editor
Animals

SI: Sheep Lactation, Nutrition and Reproduction

Comments to the revised version of:

Robles Jimenez et al.

Milk production of Lacaune sheep with different degrees of crossing with Manchega sheep in a commercial flock in Spain

Dear  Editor  and   Reviewers,

I would like to thank you for the opportunity of letting us improve and resubmit the manuscript. We believe that your suggestions have identified areas to improve that will benefit the overall presentation and content of the manuscript. Attached you will find the updated manuscript version, with the following modifications that have been applied.

Please note that although changes have been highlighted in red font, deletions of content could not be indicated.

Sincerely  the authors

Response to reviewers

L2: Change “Milk production of Lacaune sheep with different crossing degrees with Manchega sheep in a commercial flock in Spain” by “Milk production of Lacaune sheep with different degrees of crossing with Manchega sheep in a commercial flock in Spain”. Done

L22: Removed “production”. Done

L23: Change “grade” by “degree”. Done

L28: Removed “environmental resistance”. Done

L33: Change “controls” by “records”. Done

L40: Change “degree absorption increases with” by “degree of gene absorption increases with”.

L44: the effects of the number of lambing  change   by  parity number

L50: Change “The sheep” by “Sheep”. Done

L51: Removed “´s ”. Done

L51: Removed “census of ”. Done

L67: Change “sheepfold” by “barn”.

L75: Change “control” by “records”.

L75: Change “was” by “were”.

L81: Change “recods” by “records”.

L96: Removed “artificial insemination ”. Done

L100: AGRAMA is National Association of Manchega Sheep Breeders.

L105: Change “lacaune” by “Lacaune”.

L105: Removed “of ”. Done

L105: Change “drying” by “dry”.

L105: Change “/” by “per”.

L106: Change “lambing” by “lamb”.

L106: Change “simple” by “single”.

L110: Change “Due to the structure of the data of the number of lambing variables, the subset data of 8 to 13 lambings were merged to only one category called” by “Due to the structure of the data, parities from 8 to 13 were merged to only one category called >8, therefore eight levels were obtained (1 to 8th). For litter size, data of triple lambing (29 births) were grouped with the double-birth data, therefore two levels were considered: single birth (n = 4,011) and multiple (twins and triplets, n = 1,758).”.

L106: Change “double birth” by “double-birth”.

L119: Change “Statistic” by “Statistical”. Done

L20: Change “evaluated” by “evaluate”. Done

L121: Change “factor” by “factors”. Done

L132: Change “analytica” by “analytical”. Done

L136: Change “lactation to” by “lactation of ”. Done

L136: Change “analized” by “analyzed”. Done

L139: Change “Parameter ” by “parameter”. Done

L140: Change “descirbe” by “describe”. Done

L152: Change “from  the information considered  ” by “used”. Done

L153: Change “ia ” by “is ”. Done

L168: Change “Dairy” by “dairy”. Done

L139: Change “Father” by “father”. Done

L173: where and when

R: in a comercial flock in Spain.

L173: what is Drying- lambing .

R: Drying – lambing (days) is the time between the ewe drying up to the start of the next parity - lactation period

L173: Change “Milk” by “milk”. Done

L173: Change “Production” by “production”. Done

L180: Change “standard error ” by “standard error mean”. Done

L180: Change “Minimum” by “Least”. Done

L173: what this Number of lambing.

change by , Number of parity, Due to some animals lambing more than once a year, some animals have more parity records; additional parities above 8 were grouped as ≥ 8.

L190: Change “with” by “to”. Done

L190: Change “machega” by “Manchega”. Done

L213: How fitted Lactation curves.

R: Lactation curves (fited with the Wood gamma model).

L213: Change “Machega” by “Manchega”. Done

L225: Change “that” by “where”. Done

L228: Change “presents” by “showed”. Done

L229: Change “coincide” by “agree”. Done

L241: Change “finding similarity with productive” by “similar to milk yields in”. Done

L242: Change “also highlighting in this study” by “The latter study”. Done

L247: Change “it was made in a research ” by “the study was done in a research”. Done

L252-254: In relation to the lactation curves (Table 4, Figure 3), the Lacaune sheep showed a higher lactation PY (1.69; P=0.002); however, it showed a lower persistence (the maintenance of a higher level of milk production for a longer time during lactation) (4.23; P=0.001);

L261: Change “mention” by “mentioned”. Done

L262: Change “l per animal/year ” by “L per animal per year”.

L263: AGRAMA is National Association of Manchega Sheep Breeders.

L265: Change “kg /day and 120 kg / lactation at 100 days” by “kg per day and 120 kg per lactation at 100 days”

L267: Change “kg /lactation” by “kg per lactation”.

L267: Change “in Lacaune” by “in the Lacaune”.

L273: Change “of variation” by “affecting”.

L288: Change “breastfeeding ” by “during the suckling”.

L289: Change “raise” by “raised”.

L304: Change “metioning ” by “who founded”.

L304: Change “the season of the year in milk production were different” by “These results are similar to those of Rovai et al. [17], who also found differences in milk production depending on the season of the year. However, Rovani et al. [17] found lower milk yields in winter, while our study did not see lower yields in winter, but instead saw lower yields in summer and autumn”

L309: Removed “production”. Done

L310: Change “decreases” by “decreased”.

L311: Change “crosse ” by “crosses”.

L395: was added Peterson, S.W; Mackenzie, D.D.S.; McCutcheon, S.N. Milk production and plasma prolactin levels in spring-and autumn-lambing ewes. Proc. N. Z. Soc. Anim. Prod. 1990. 50, 483-485
